# Trimester-Specific Serum Fructosamine in Association with Abdominal Adiposity, Insulin Resistance, and Inflammation in Healthy Pregnant Individuals

**DOI:** 10.3390/nu14193999

**Published:** 2022-09-27

**Authors:** Emilie Bernier, Amélie Lachance, Anne-Sophie Plante, Patricia Lemieux, Karim Mourabit Amari, S. John Weisnagel, Claudia Gagnon, Andréanne Michaud, André Tchernof, Anne-Sophie Morisset

**Affiliations:** 1École de Nutrition, l’Université Laval, Québec, QC G1V 0A6, Canada; 2Centre Nutrition, Santé et Société (NUTRISS), l’Institut sur la Nutrition et les Aliments Fonctionnels (INAF), l’Université Laval, Québec, QC G1V 0A6, Canada; 3Axe Endocrinologie et Néphrologie, Centre de Recherche, CHU de Québec-Université Laval, Québec, QC G1V 4G2, Canada; 4Centre de Recherche de l’Institut Universitaire de Cardiologie et de Pneumologie, Québec-Université Laval, Québec, QC G1V 4G5, Canada; 5Département de Médecine, l’Université Laval, Québec, QC G1V 0A6, Canada; 6Département de Médecine de Laboratoire, CHU de Québec-Université Laval, Québec, QC G1V 4G5, Canada

**Keywords:** pregnancy, fructosamine, adiposity, body fat distribution, glucose homeostasis, insulin resistance, gestational diabetes mellitus, inflammation

## Abstract

This study aimed to (1) characterize the variations in serum fructosamine across trimesters and according to pre-pregnancy BMI (ppBMI), and (2) examine associations between fructosamine and adiposity/metabolic markers (ppBMI, first-trimester adiposity, leptin, glucose homeostasis, and inflammation measurements) during pregnancy. Serum fructosamine, albumin, fasting glucose and insulin, leptin, adiponectin, interleukin-6 (IL-6), and C-reactive protein (CRP) concentrations were measured at each trimester. In the first trimester, subcutaneous (SAT) and visceral (VAT) adipose tissue thicknesses were estimated by ultrasound. In the 101 healthy pregnant individuals included (age: 32.2 ± 3.5 y.o.; ppBMI: 25.5 ± 5.5 kg/m^2^), fructosamine concentrations decreased during pregnancy whereas albumin-corrected fructosamine concentrations increased (*p* < 0.0001 for both). Notably, fructosamine concentrations were inversely associated with ppBMI, first-trimester SAT, VAT, and leptin (r = −0.55, r = −0.61, r = −0.48, r = −0.47, respectively; *p* < 0.0001 for all), first-trimester fasting insulin and HOMA-IR (r = −0.46, r = −0.46; *p* < 0.0001 for both), and first-trimester IL-6 (r = −0.38, *p* < 0.01). However, once corrected for albumin, most of the correlations lost strength. Once adjusted for ppBMI, fructosamine concentrations were positively associated with third-trimester fasting glucose and CRP (r = 0.24, r = 0.27; *p* < 0.05 for both). In conclusion, serum fructosamine is inversely associated with adiposity before and during pregnancy, with markers of glucose homeostasis and inflammation, but the latter associations are partially influenced by albumin concentrations and ppBMI.

## 1. Introduction

Gestational diabetes mellitus (GDM) is one of the most frequent health conditions affecting pregnant individuals [1]. This could partially be brought on by the fact that an increasing number of individuals enter pregnancy being overweight or obese, with increased adipose fat storage. Indeed, independently of other known risk factors for GDM, ultrasound-estimated maternal visceral adipose tissue (VAT) thickness in pregnancy is strongly associated with insulin resistance and higher risks of developing GDM [2,3,4,5]. Along with increased adiposity, GDM can have detrimental health consequences for both the mother and the fetus [6,7,8,9]. Hence, glycemic control is particularly fundamental in all pregnant individuals (with or without GDM), as continuous associations between maternal glucose levels (below those of GDM diagnosis) and adverse pregnancy outcomes, such as primary cesarean delivery and neonatal hypoglycemia, have been observed [10]. Several risk factors have been implicated in the development of glucose intolerance and GDM, with maternal weight status being the most commonly evaluated reversible risk factor [9,11].

In individuals with type 1 or type 2 diabetes, glycated hemoglobin (HbA1C) is the recommended marker to monitor long-term glycemic variations, as it reflects the average glycemia level of the past three months [12]. However, this marker has some important limitations within the gestational context since it is affected by the lifespan of erythrocytes [13], which are known to be decreased by natural pregnancy-related anemia and iron deficiency [14,15,16]. Furthermore, because of the rapid changes in glycemic control and insulin sensitivity that may occur over this period [15], a shorter-term marker for blood glucose homeostasis is needed. The current literature suggests that serum fructosamine could be an alternative to follow glycemic variations [17], as it reflects glycemic variations of the last two to three weeks approximately [14,15,18,19]. Fructosamine is a sugar-protein complex that forms in chronic hyperglycemic conditions. Since albumin is the most abundant protein in the blood, fructosamine levels typically reflect albumin glycation [20,21]. Its measurement is rapid, inexpensive, precise, and simple [15,18,22]. Unlike HbA1c, it is not impacted by conditions affecting red blood cell concentration, such as pregnancy, hemoglobinopathies, or hemolytic anemia [15,18,22,23]. Nevertheless, conditions that affect serum protein and albumin metabolism can also influence fructosamine levels, as fructosamine levels correlate significantly with total protein and albumin concentrations [15,24]. Some methods are recommended in the literature to correct fructosamine levels based on total circulating protein or albumin, as corrected fructosamine is shown to better correlate with glucose homeostasis markers and GDM diagnosis [24]. Since albumin concentrations are known to decrease during pregnancy due to dilution [25,26] and to the natural low-grade inflammation developing over trimesters, correction for albumin concentrations could reverse the results observed with uncorrected fructosamine levels, as seen by others [27]. Although, there is a debate about whether fructosamine concentration should be corrected for total albumin, total protein, or neither, and best practices have not been established [28]. Therefore, in this study, we use both uncorrected and albumin-corrected fructosamine levels, thus enabling comparison with other studies. At present, clinical fructosamine monitoring is used only among populations where HbA1c is thought to inaccurately reflect glycemia, such as hemoglobinopathies and severe kidney disease [15,29,30], but it is not frequently used in pregnancy. Still, a few studies have evaluated the accuracy of fructosamine to monitor glycemia in pregnancy or to predict GDM and reported contrasting results [22,25,31,32,33,34,35].

On the one hand, hyperglycemia has been proposed to cause protein glycation, which can induce the activation of several intracellular pathways and the release of pro-inflammatory mediators [36,37,38]. Thus, by the formation of fructosamine, glycemic imbalance could lead to inflammation. On the other hand, pro-inflammatory cytokines such as interleukin-6 (IL-6) have been suggested to contribute to the overall development of insulin resistance [39] and could have a role in the etiology of GDM [40]. Even though the links between abdominal fat accumulation, adipose tissue dysfunction, adiposity-related systemic inflammation, and insulin resistance have been well demonstrated [41,42], no study has ever investigated the associations between fructosamine concentrations and abdominal adiposity or inflammation in pregnancy. Furthermore, even if BMI has been shown to strongly correlate with fructosamine concentrations [43,44,45] its confounding effects have never been studied in those associations. Thus, this study aims to (1) characterize the variations of fructosamine across trimesters and according to pre-pregnancy BMI (ppBMI) categories and (2) examine associations between fructosamine concentrations (both uncorrected and corrected for albumin concentrations) and (a) adiposity (ppBMI, first-trimester subcutaneous (SAT) and visceral (VAT) adipose thicknesses, and leptin), (b) glucose homeostasis (fasting blood glucose, fasting blood insulin, and an insulin resistance index (HOMA-IR)) and (c) inflammation measurements (adiponectin, IL-6, and C-reactive protein (CRP) in each trimester, with and without adjustment for ppBMI. The authors hypothesize that (1) serum uncorrected fructosamine levels decrease during pregnancy, whereas albumin-corrected ones increase, and (2) both uncorrected and albumin-corrected fructosamine concentrations are associated positively with adiposity, glucose homeostasis, and inflammatory measurements.

## 2. Materials and Methods

### 2.1. Study Population

This is an exploratory analysis of data collected prospectively in 101 pregnant individuals from the ANGE (Apports Nutritionnels durant la GrossessE) and PAGG (Prise alimentaire, Apports nutritionnels et Gain de poids durant la Grossesse) cohort studies. The ANGE cohort is composed of 86 healthy pregnant individuals who were recruited between April 2016 and May 2017 at the Centre de recherche du CHU de Québec-Université Laval in Quebec City, Canada, in their first trimester and who were followed until delivery. The study objectives and protocol were previously described elsewhere [46,47]. The present analysis includes data from 78 pregnant individuals from this cohort, as seven women were lost at follow-up, due to miscarriage (*n* = 3) or lack of time to devote to the study (*n* = 4), and one had incomplete fructosamine data.

The PAGG cohort consists of 33 pregnant individuals who were recruited in their first trimester, as part of a pilot project between November 2018 and September 2020 also at the CHU de Québec-Université Laval, and who were followed until delivery. The aim of the PAGG study, for which data analysis is still ongoing, is to evaluate food-related hormones variations across pregnancy and to analyze their associations with trimester-specific dietary intakes and gestational weight gain. Our final analysis includes 23 pregnant individuals from this cohort since one was excluded for having a multiple pregnancy, four were lost at follow-up due to miscarriage (*n* = 1), premature birth (*n* = 1), or reluctancy to come to on-site visits because of COVID-19 (SARS-CoV-2) (*n* = 2), and three were excluded for the present analysis because of incomplete fructosamine data. Finally, two PAGG participants who had already participated in the ANGE project were excluded from the PAGG analyzed sample.

The selection criteria were identical for both cohorts. Individuals of 18 years old or more who had completed 11 or fewer gestational weeks at the time of recruitment and who were able to consent were eligible for the study. Individuals with multiple pregnancy or with a previously diagnosed severe illness that may influence their metabolic status or intakes (i.e., type 1 and type 2 diabetes, renal disease, immunity, and inflammatory disorders) were excluded. Both study protocols were similar, as they included on-site visits at each trimester, with metabolic testing, using the exact same methods, as well as the same self-administered Web questionnaires, as previously detailed elsewhere [46,47]. Both projects followed the Declaration of Helsinki guidelines and were approved by the Ethics Committee of the Centre de recherche du CHU de Québec-Université Laval (ANGE reference number: 2016-2866; PAGG reference number: 2019-4342). All participants gave their informed written consent at their first on-site visit.

### 2.2. Measurements of Fructosamine and Albumin-Corrected Fructosamine

Blood samples were taken at each trimester (12.5 ± 0.7, 22.8 ± 1.0, and 33.6 ± 1.3 gestational weeks), after a 12 h fast. They were then centrifuged for serum and stored at −80 °C for future analyses.

Fructosamine was measured by the nitroblue tetrazolium coloration assay (Roche, Cobas C, Indianapolis, IN, USA). Participants who had one or more hemolyzed samples (>2.5 g/L hemoglobin), which could affect the reliability of the results, were excluded from the present analysis (*n* = 13). Serum albumin was measured using the bromocresol purple dye-binding assay (Siemens, Dimension Vista System, ALB, Newark, DE, USA). Since the correction of fructosamine for total protein concentration or albumin is recommended in variables states of hydremia such as pregnancy [48], fructosamine concentrations were used both uncorrected and corrected for albumin using the following correction formula [49]:Albumin-corrected fructosamine = (Fructosamine concentration [μmol/L]/albumin concentration [g/L]) × 30),(1)
the mean serum concentrations of albumin in the whole data set being 30, as did other groups [24,27].

### 2.3. Measurements of Adiposity

A subset of 95 participants underwent ultrasound measurement of subcutaneous (SAT) and visceral (VAT) adipose tissue thicknesses in their first trimester on a Voluson E8 Expert system (GE Healthcare Inc., Milwaukee, IL, USA) ultrasound machine using a RAB4-8-D/OB probe. Qualified ultrasound technicians employed a previously reported method to do transabdominal ultrasonography [50]. Briefly, measurements were taken with participants in the supine position, with the probe perpendicular to the aorta, at the level of the Linea alba (2.5 cm above the umbilicus). Images were taken after an expiration when the aorta appeared to be closest to the surface. SAT thickness was measured from the subcutaneous fat layer to the outer border of the rectus abdominis muscle at the level of the Linea alba, whereas VAT thickness was measured from the inner border of the rectus abdominis muscle at the level of the Linea alba to the anterior wall of the abdominal aorta. Weeks of pregnancy were also confirmed by ultrasound at the same visit.

Trimester-specific levels of leptin (EMD Millipore Corporation, Billerica, MA, USA) were measured by ELISA kits. This adiposity marker was only available for the ANGE cohort (*n* = 78). Data from participants with extreme values with more than 2 standard deviations (SD) from the mean were excluded. Only complete data (available at each trimester) were included in the current study (*n* = 74).

### 2.4. Measurements of Glucose Homeostasis

From fasting blood samples drawn at every trimester, glucose was measured enzymatically by the hexokinase method (Siemens, Dimension Vista 1500, CV 1.8%, Newark, DE, USA), whereas insulin was measured with an electrochemiluminescence immunoassay (Siemens, Advia Centaur XPT, CV 5%, Newark, DE, USA). The present analyses were performed with data of participants from which we had available and valid measurements at every trimester (*n* = 88, both for glucose and insulin). Insulin resistance was estimated using the Homeostasis Model Assessment of Insulin Resistance (HOMA-IR) [51]:HOMA-IR = (Fasting insulin [μU/mL] × fasting glucose [mmol/L])/22.5).(2)

### 2.5. Measurements of Inflammation

Trimester-specific levels of adiponectin (B-Bridge International Inc, Santa Clara, CA, USA), and IL-6 (R&D Systems Inc., Minneapolis, MN, USA) were measured by ELISA kits, whereas C-reactive protein (CRP) concentrations were obtained using a high sensitivity immunonephelometry test (Siemens Medical Solutions USA Inc., Malvern, PA, USA). Those markers were only available for the ANGE cohort (*n* = 78). Data from participants with extreme values with more than 2 SD from the mean were excluded. Only complete data (available at each trimester) were included in the current study, which comprised a total of 72 samples for adiponectin, and 70 for both IL-6 and CRP.

### 2.6. Other Variables

PpBMI (weight (kg)/height (m)^2^) was calculated using self-reported pre-pregnancy weight and on-site measured height and categorization was as follows for analyses: underweight and normal weight (<25 kg/m^2^), overweight (25.0–29.9 kg/m^2^), or obese (≥30.0 kg/m^2^). Information on ethnicity, education level, and annual household income were collected by self-administered Web-based questionnaires. Diagnosis of GDM, made according to the guidelines in effect at the time of the test [11,52], was collected directly from the participant’s medical records.

### 2.7. Statistical Analyses

Means, SD, and proportions were used to characterize the study sample. Mixed models for repeated measures analysis of variance were conducted to examine the evolution of circulating concentrations of fructosamine and albumin-corrected fructosamine across trimesters, and according to ppBMI. Participants’ identification number was included as a random effect and both trimesters of pregnancy and ppBMI were included as fixed effects. If the mixed model residuals were not normally distributed, the dependent variable value for each participant was transformed using a Box-Cox transformation. Post hoc comparisons for mixed models were carried out using Tukey–Kramer honest significant difference (HSD) tests. When the change in concentrations across trimesters was significant, the test slice analysis (repeated measures mixed model for each ppBMI category) was performed to identify in which category the change in concentrations was significant. Analysis of variance (ANOVA) with Tukey–Kramer HSD posthoc tests were then used to examine differences in concentrations of fructosamine and albumin-corrected fructosamine between ppBMI categories for each trimester separately. To examine the associations of fructosamine and albumin-corrected fructosamine concentrations with ppBMI, adiposity, glucose homeostasis, and inflammation measurements, Spearman’s correlations (non-linear association of variables) or Pearson’s correlations (linear association of variables) were performed, as appropriate based on the association observed on plots. Correlation coefficients are presented as both raw and adjusted for ppBMI. ANOVA were used to examine the differences in fructosamine concentrations between participants with and without GDM. All statistical analyses were performed with JMP Pro version 15.2 (SAS Institute Inc., Cary, NC, USA). The threshold of statistical significance was defined at *p* ≤ 0.05, and 0.05 < *p* ≤ 0.1 were considered as trends only.

## 3. Results

### 3.1. Participant Characteristics

Participant characteristics are presented in Table 1. The final analysis was conducted on a total of 101 pregnant individuals recruited at 10.1 ± 1.6 weeks of gestation. Participants were aged 32.2 ± 3.5 years at the time of recruitment and had a mean ppBMI of 25.5 ± 5.5 kg/m^2^. Most of them were categorized as having a normal weight (54.5%), while three participants (2.9%) were categorized as being underweight, 25 (24.8%) as overweight, and 18 (17.8%) as obese. Most participants were of European descent (97.0%), multiparous (63.4%), university degree holders (77.2%), and had an annual household income of 80,000 Canadian dollars or above (65.3%). Eleven individuals (10.9%) were diagnosed with GDM during their pregnancy.

### 3.2. Variations in Fructosamine and Albumin-Corrected Fructosamine throughout Pregnancy

Serum fructosamine levels decreased significantly between the first, second, and third trimesters of pregnancy (*p* < 0.0001) (Figure 1A). Concentrations of fructosamine decreased significantly for all ppBMI categories (test slice *p*-values < 0.0001, <0.0001, 0.0002, for each ppBMI category, respectively). Circulating fructosamine was significantly lower in every trimester in individuals with a ppBMI ≥ 30 kg/m^2^, compared to those with a ppBMI < 25 kg/m^2^ (Tukey–Kramer HSD *p*-values < 0.0001, <0.009, <0.03, for each trimester, respectively). Individuals with a ppBMI ≥ 30 kg/m^2^ had lower fructosamine values compared to those with a 25 < ppBMI < 30 kg/m^2^ in the first trimester only (Tukey–Kramer HSD *p*-value < 0.0002).

After correction for albumin, fructosamine concentrations increased throughout pregnancy (*p* < 0.0001), with significant increases between the first and second trimesters and between the first and third trimesters (*p* < 0.0001 for both) (Figure 1B). Concentrations of albumin-corrected fructosamine increased significantly throughout trimesters for all ppBMI categories (test slice *p*-values < 0.0001, 0.002, 0.0002, for each ppBMI category, respectively). Even though the evolution of fructosamine concentrations tended to be different across ppBMI categories (*p*-value = 0.09), no differences in albumin-corrected fructosamine concentrations were significant between ppBMI categories, regardless of the trimester (Figure 1B).

### 3.3. Associations between Fructosamine and Albumin-Corrected Fructosamine and Adiposity Measurements

As shown in Table 2, first-trimester fructosamine concentrations were strongly and inversely correlated with ppBMI. Even though the correlations lost strength as the pregnancy progressed, they remained significant for both the second and third trimesters. Inverse correlations were also observed between fructosamine concentrations and first-trimester ultrasound-estimated SAT and VAT thickness, and leptin concentrations (Table 2). However, all associations for SAT and VAT became nonsignificant after adjustment for ppBMI. Once corrected for albumin, the strength of the latter correlations decreased but remained significant, except for leptin (Table 3). First-trimester serum albumin was inversely associated with ppBMI, SAT, and leptin (Table 4).

### 3.4. Associations between Fructosamine and Albumin-Corrected Fructosamine and Glucose Homeostasis Measurements

Many inverse associations between fructosamine, fasting insulin, and HOMA-IR were found in the first and third trimesters (Table 2). Nevertheless, all associations lost significance after adjustment for ppBMI. Few associations emerged through ppBMI adjustment, as fructosamine was positively associated with second- and third-trimester fasting glucose. After correction for albumin, only first-trimester fasting insulin, and HOMA-IR were found to correlate inversely with fructosamine levels (Table 3), but lost some strength. As shown in Table 4, serum albumin concentrations were inversely associated with first-trimester fasting insulin and a trend was also observed for HOMA-IR. Although, the latter associations did not remain significant after ppBMI adjustment. Additional exploratory analyses have shown that no statistical difference was significant between uncorrected fructosamine and albumin-corrected fructosamine concentrations of participants with (*n* = 11) versus those without GDM (*n* = 90).

### 3.5. Associations between Fructosamine and Albumin-Corrected Fructosamine and Inflammation Measurements

As for inflammation measurements, adiponectin and IL-6 were, respectively correlated positively and inversely with fructosamine only in the first trimester (Table 2). Those associations did not remain significant with the adjustment for ppBMI, as only a trend remained for adiponectin. Independently of ppBMI, CRP was found to be positively associated with fructosamine in the third trimester. When analyses were made with albumin-corrected fructosamine and adjusted for ppBMI, CRP and IL-6 were positively associated with fructosamine levels in the first and second trimesters, respectively (Table 3). As shown in Table 4, serum albumin concentrations were inversely and strongly associated with second- and third-trimester IL-6 levels, as well as first-trimester CRP levels. Albumin levels were also positively associated with second-trimester adiponectin levels.

## 4. Discussion

As hypothesized, fructosamine concentrations decreased significantly during pregnancy, while the opposite was observed after correcting for albumin. Adiposity markers (ppBMI, SAT, VAT, and leptin), as well as measurements of glucose homeostasis (fasting insulin and HOMA-IR), were inversely correlated with fructosamine. These associations were stronger in the first trimester and the majority was lost after adjustments for ppBMI. However, independently of this covariate, fructosamine concentrations were positively associated with fasting glucose and CRP levels in late pregnancy, and albumin-corrected fructosamine concentrations were positively associated with CRP and IL-6 in early and mid-pregnancy.

These findings suggest that fructosamine concentrations decrease throughout pregnancy and this confirms the findings of other groups [25,27,35,48]. The increase in fructosamine concentrations observed after correction for albumin has also already been reported in both Caucasian and Asian non-diabetic populations [27]. Such results highlight the need to consider variations in protein concentration when using fructosamine as a gestational marker and suggest that the decrease in uncorrected fructosamine concentrations might reflect the natural pregnancy-related decrease in albumin concentrations.

Contrary to the initial hypothesis, pregnant individuals with higher adiposity measurements before (ppBMI) and in early pregnancy (SAT and VAT thicknesses) appeared to have lower uncorrected fructosamine concentrations. All associations observed between fructosamine and adipose tissue thicknesses were partially influenced by ppBMI. Other studies in the general population demonstrated inverse associations between uncorrected fructosamine and BMI, both in diabetic and non-diabetic samples [43,44,45]. Contrary to our results, a study investigating body fat distribution and glycemic control among Maya women in rural Yucatán demonstrated that variables reflecting overall adiposity and central adiposity were positively associated with fructosamine [53]. However, this study was conducted on a small sample (*n* = 60), with adiposity being only measured with anthropometric variables, with no consideration for albumin correction. Other studies in the general population have demonstrated that serum fructosamine levels were associated negatively with fat mass [43], but no association was observed with waist circumference, fat mass, liver fat, and fat distribution [39,43,54]. Therefore, the literature generally shows contrasting results regarding the associations between fructosamine and adiposity. The inverse correlations between total albumin and adiposity could influence this associations. As proposed by others, in obese individuals exhibiting chronic inflammation, the pro-inflammatory cytokines could reduce albumin synthesis and speed up albumin catabolism, thus resulting in lower fructosamine levels [41,55].

In the first and third trimesters, several inverse associations between fructosamine, fasting insulin, and HOMA-IR were observed. A Canadian group investigating the associations of second-trimester fructosamine levels with 3-year postpartum glycemic indices in GDM and non-GDM participants has demonstrated that higher fructosamine levels were associated with higher maternal 3-year postpartum fasting insulin and HOMA-IR [22], whereas no associations were observed between the fructosamine levels and HOMA-IR in a sample of patients with type 2 diabetes [56]. Nevertheless, since all correlations lost their significance after accounting for ppBMI, uncorrected fructosamine concentrations may be more associated with markers of overall adiposity (ppBMI, SAT) than with insulin levels and insulin resistance itself. Although, independently of ppBMI, second- and third-trimester fasting glucose concentrations were positively associated with fructosamine, confirming previously published data collected among pregnant [22,25,34] and non-pregnant populations [44,57,58]. The positive association between fructosamine and fasting blood glucose could be explained by the glycation mechanisms leading to the formation of fructosamine. As more glucose is available in circulation, more glucose will covalently be attached to serum proteins and more fructosamine will be formed. Those correlations emerged only after adjustment for ppBMI, which could be explained by the fact that ppBMI influences this association, as previously seen by a group of non-pregnant participants [57]. After accounting for albumin, the significance of the observed associations was lost, which has already been observed in overweight and obese animal models [59]. Finally, it should be noted that much weaker correlations were observed between fasting glucose and fructosamine levels than between markers of insulin resistance and fructosamine levels. Those findings are consistent with evidence suggesting that fructosamine concentrations are a poor predictor of gestational glucose tolerance and risk of developing GDM [31,34,35].

No significant differences were observed in both uncorrected and albumin-corrected fructosamine concentrations between participants with and without GDM. Although, this result should be interpreted with caution because only 11 individuals in our sample developed GDM during follow-up, which may not give us the statistical power to detect differences between groups. This, in addition to the very few significant associations observed between fructosamine concentrations and glycemic markers, lead us to believe that, in a population with generally adequate glycemic control, serum fructosamine does not appear to be a marker associated with the development of GDM. However, in a diabetic population, different results could have been found, as demonstrated by other studies which suggested that fructosamine concentrations were higher in pregnant individuals with GDM or with glucose abnormalities [31,35]. Although, these aspects remain to be explored since the present sample did not allow to answer this precise question.

Some inverse associations were also noticed between fructosamine concentrations and inflammation (adiponectin and IL-6) throughout pregnancy, without being independent of ppBMI. A cross-sectional analysis conducted in the Pakistani general population, aiming to investigate the correlations between glycated proteins and periodontal and systemic inflammation also demonstrated inverse associations, as sixteen inflammatory proteins were found to correlate inversely with fructosamine [39]. The inverse associations observed between uncorrected fructosamine and inflammatory measurements could reflect the negative associations between albumin and inflammation, as all significant correlations were lost after albumin correction. Once corrected for albumin concentrations, fructosamine appeared to be positively associated with inflammation. Another group recently demonstrated that fructosamine was significantly increased in non-pregnant diabetic patients with acute infection compared to those without infection, with a positive correlation between the parameters of inflammation and glucoregulation [60]. Those results are supported by an in vitro study demonstrating that glycated albumin stimulates the expression of several proinflammatory cytokines [61]. We can speculate that hyperglycemia can lead to protein glycation, which can then trigger the release of pro-inflammatory mediators [36,37,38]. Inversely, it is also possible that proinflammatory cytokines, originating from either adiposity or pregnancy-related inflammation, could contribute to the general development of insulin resistance and protein glycation [39].

The results observed in the current study suggest that the evolution of fructosamine concentrations, as well as the associations found between fructosamine concentrations and metabolic measurements, may be mediated by albumin concentrations. Hence, the associations found between serum albumin concentrations and the studied metabolic markers could explain, in part, why the observed associations between serum fructosamine concentrations and the studied markers lost strength after correction for albumin. Furthermore, serum albumin has already been inversely associated with adipose tissue inflammation, adiposity, and glucose regulation, and could even predict the risk of developing type 2 diabetes in a longitudinal cohort of healthy adults [62]. We propose that increased adiposity, reflected as higher ppBMI and thicker abdominal adipose tissues in early pregnancy, may result in adipose tissue dysfunction and adiposity-related systemic inflammation [41]. In turn, this may decrease albumin synthesis and accelerate albumin catabolism, resulting in lower protein glycation, as seen by lower fructosamine concentrations [55]. Moreover, knowing that a higher ppBMI is associated with lower fructosamine concentrations, the associations between ppBMI and metabolic measurements could be stronger than the associations between fructosamine and metabolic measurements, resulting in the loss of statistical significance in most of our associations with the adjustment for this covariate. Still, independently of ppBMI, some positive associations remained between fructosamine and fasting glucose, IL-6, and CRP, mainly in the second half of pregnancy. Thus, with advancing pregnancy, the natural development of insulin resistance and low-grade inflammation could promote hyperglycemia, which could potentiate fructosamine formation through protein glycation and thereby contribute to the development and maintenance of a pro-inflammatory state. Hence, the influence of ppBMI on the glycemic profile and inflammation could lose its power with the advancing of pregnancy, where other physiologic factors may come into play.

To the best of our knowledge, this is the first study to examine the associations between fructosamine concentrations and a wide diversity of metabolic measurements. It is also one of the few studies to investigate these associations in a group of healthy participants, as most available data are from diabetic patients. The present study has various strengths, namely the very early recruitment in pregnancy that allowed the measurement of studied markers at three time points and the measurement of three distinct inflammatory markers. The reproduction of all analyses, with and without correction for albumin, also allowed us to demonstrate the importance of this correction in the gestational context. Nevertheless, since the sample size was small and was predominantly composed of pregnant individuals of European ancestry from high socioeconomic standing, this study needs to be replicated in more diversified populations. Due to the small number of subjects who acquired GDM, we were unable to detect significant differences in fructosamine concentrations when compared to non-GDM pregnant individuals. Although, since even mildly elevated glucose and insulin levels are known to have negative effects on the health of the mother and child even below the GDM diagnosis, the chosen glucose homeostasis markers seemed to be appropriate and allowed us to precisely characterize the nature of the associations between fructosamine and glucose regulation during a healthy pregnancy. Finally, it is to be noted that no information on gender identification was available, making it impossible to assume that all study subjects identified as women.

## 5. Conclusions

In conclusion, the gradual decrease in circulating fructosamine and increase in albumin-corrected fructosamine concentrations are in line with the physiological decrease in albumin due to dilution and inflammation settling during pregnancy, but other studies are necessary to understand how ppBMI influences their evolution. These findings demonstrate that fructosamine, but mostly albumin concentrations, could come into play in the associations between adiposity, insulin resistance, and inflammation in the gestational context. Additionally, more prospective studies are required to fully understand the associations between fructosamine, albumin, insulin resistance, and inflammation throughout pregnancy. Still, fructosamine appeared to be linked with adiposity before and during early pregnancy, as well as glycemia and inflammation, but most of the associations were partially influenced by ppBMI and albumin concentrations.

## Figures and Tables

**Figure 1 nutrients-14-03999-f001:**
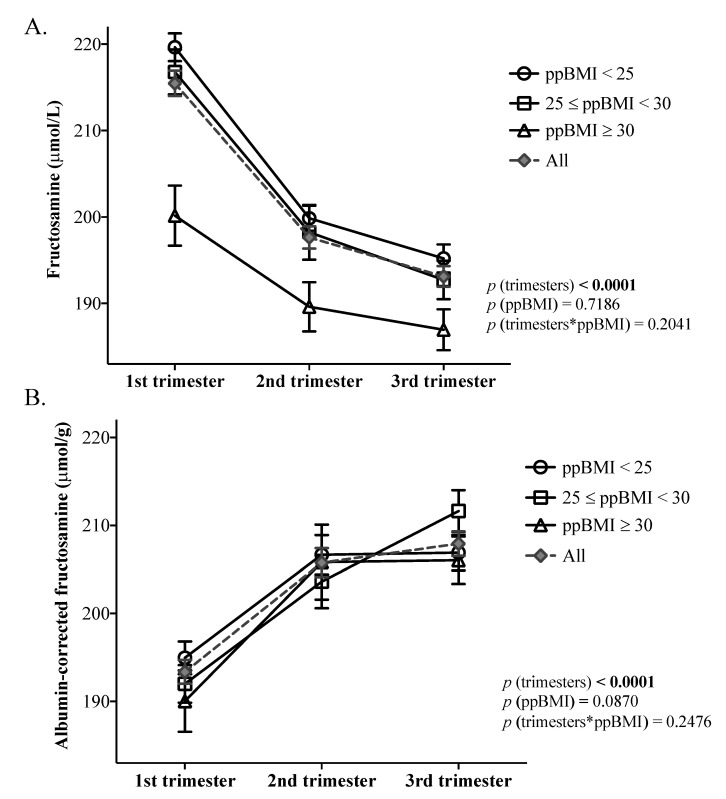
Circulating fructosamine and albumin-corrected fructosamine concentrations across trimesters, according to pre-pregnancy BMI categories. Legend. *p*-values refer to the mixed models for repeated measures used to examine the evolution of trimester-specific circulating concentrations of (**A**) fructosamine and (**B**) albumin-corrected fructosamine according to ppBMI. ppBMI, pre-pregnancy body mass index. *n* = 101.

**Table 1 nutrients-14-03999-t001:** Participants characteristics (*n* = 101).

Variables	Mean ± SD or *n* (%)
Age, years	32.2 ± 3.5
Parity ^a^	
0	37 (36.6)
≥1	64 (63.4)
Pre-pregnancy body mass index, kg/m^2^	25.5 ± 5.5
Underweight	3 (2.9)
Normal weight	55 (54.5)
Overweight	25 (24.8)
Obesity	18 (17.8)
Gestational diabetes mellitus	11 (10.9)
Ethnicity	
European descent	98 (97.0)
African	1 (1.0)
Latin American	2 (2.0)
Education	
High school	5 (5.0)
College ^b^	18 (17.8)
University	78 (77.2)
Annual household income	
<60,000 CAD	20 (19.8)
60,000–79,999 CAD	14 (13.9)
80,000–99,999 CAD	22 (21.8)
≥100,000 CAD	44 (43.6)
Income not disclosed	1 (0.9)
Fructosamine (μmol/L)	
First trimester	215.5 ± 14.7
Second trimester	197.6 ± 13.0
Third trimester	193.1 ± 12.0
Albumin (g/L)	
First trimester	33.5 ± 2.3
Second trimester	28.9 ± 2.1
Third trimester	27.9 ± 1.9
Albumin-corrected fructosamine (μmol/g)	
First trimester	193.4 ± 13.7
Second trimester	205.8 ± 16.8
Third trimester	207.9 ± 14.0
Ultrasound-estimated SAT thickness (mm) ^c^	17.4 ± 7.7
Ultrasound-estimated VAT thickness (mm) ^c^	27.0 ± 14.7
Leptin (ng/mL) ^d^	
First trimester	33.6 ± 20.2
Second trimester	38.9 ± 27.6
Third trimester	41.2 ± 26.8
Fasting glucose (mmol/L)	
First trimester ^e^	4.5 ± 0.3
Second trimester ^f^	4.4 ± 0.4
Third trimester ^f^	4.6 ± 0.5
Fasting insulin (pmol/L)	
First trimester ^g^	63.1 ± 34.0
Second trimester ^h^	74.8 ± 41.4
Third trimester ^c^	97.5 ± 47.9
HOMA-IR	
First trimester ^g^	1.8 ± 1.1
Second trimester ^h^	2.2 ± 1.4
Third trimester ^c^	2.9 ± 1.6
Adiponectin (mg/mL) ^d^	
First trimester	10.2 ± 3.2
Second trimester	9.2 ± 3.2
Third trimester	7.7 ± 2.7
IL-6 (pg/mL) ^i^	
First trimester	1.0 ± 0.6
Second trimester	1.2 ± 0.9
Third trimester	1.5 ± 0.9
CRP(mg/L) ^j^	
First trimester	7.0 ± 7.4
Second trimester	6.5 ± 5.6
Third trimester	6.5 ± 6.0

^a^ Refers to the number of children previously born, excluding the current pregnancy. ^b^ In Québec, college refers to a degree obtained after high school and before university. ^c^
*n* = 95. ^d^
*n* = 75. ^e^
*n* = 101. ^f^
*n* = 100. ^g^
*n* = 97. ^h^
*n* = 96. ^i^
*n* = 72. ^j^
*n* = 73. SAT, subcutaneous adipose tissue; VAT, visceral adipose tissue; HOMA-IR, homeostasis model assessment of insulin resistance; IL-6, interleukin-6; CRP, C-reactive protein. Longitudinal raw data for every metabolic marker according to ppBMI categories are available in Appendix A.

**Table 2 nutrients-14-03999-t002:** Trimester-specific associations between circulating uncorrected fructosamine and metabolic measurements.

		First Trimester		Second Trimester		Third Trimester	
	*n*	r	r_adj_	r	r_adj_	r	r_adj_
Adiposity markers							
ppBMI	101	−0.55 ***	-	-	-	-	-
SAT thickness	84	−0.61 ***	−0.16	-	-	-	-
VAT thickness	84	−0.48 ***	−0.03	-	-	-	-
Leptin	74	−0.47 ***	−0.003	−0.26 *	−0.08	−0.25 *	−0.01
Glucose homeostasis markers							
Fasting glucose	88	−0.04	0.16	0.16	0.24 *	0.10	0.24 *
Fasting insulin	88	−0.46 ***	0.01	−0.16	−0.002	−0.27 *	0.01
HOMA-IR	88	−0.46 ***	0.05	−0.10	0.06	−0.22 *	0.05
Inflammatory markers							
Adiponectin	72	0.33 **	0.16	0.23 ^†^	0.13	0.13	0.08
IL-6	70	−0.38 **	−0.07	−0.05	0.05	0.06	0.09
CRP	70	−0.19	0.08	−0.07	0.09	0.10	0.27 *

Values are presented as Pearson or Spearman correlation coefficients, raw (r) or adjusted for ppBMI (radj). *** *p* < 0.0001, ** *p* < 0.01, * *p* < 0.05, ^†^
*p* < 0.1. ppBMI, pre-pregnancy body mass index; SAT, subcutaneous adipose tissue; VAT, visceral adipose tissue; HOMA-IR, homeostasis model assessment of insulin resistance; IL-6, interleukin-6; CRP, C-reactive protein.

**Table 3 nutrients-14-03999-t003:** Trimester-specific associations between albumin-corrected fructosamine and metabolic measurements.

		First Trimester		Second Trimester		Third Trimester	
	*n*	r	r_adj_	r	r_adj_	r	r_adj_
Adiposity markers							
ppBMI	101	−0.20 *	-	-	-	-	-
SAT thickness	84	−0.30 **	−0.15	-	-	-	-
VAT thickness	84	−0.30 **	−0.17	-	-	-	-
Leptin	74	−0.15	−0.02	−0.06	0.19	0.02	0.09
Glucose homeostasis markers							
Fasting glucose	88	−0.09	−0.04	0.02	−0.04	0.12	0.15
Fasting insulin	88	−0.23 *	−0.09	−0.02	0.10	−0.003	0.05
HOMA-IR	88	−0.22 *	−0.09	−0.001	0.10	−0.004	0.05
Inflammatory markers							
Adiponectin	72	0.22 ^†^	0.16	−0.02	−0.05	−0.03	−0.04
IL-6	70	−0.17	−0.10	0.29 *	0.31 **	0.06	0.06
CRP	70	0.20 ^†^	0.27 *	−0.05	−0.03	0.06	0.09

Values are presented as Pearson or Spearman correlation coefficients, raw (r) or adjusted for ppBMI (radj). ** *p* < 0.01, * *p* < 0.05, ^†^
*p* < 0.1. ppBMI, pre-pregnancy body mass index; SAT, subcutaneous adipose tissue; VAT, visceral adipose tissue; HOMA-IR, homeostasis model assessment of insulin resistance; IL-6, interleukin-6; CRP, C-reactive protein.

**Table 4 nutrients-14-03999-t004:** Trimester-specific associations between albumin and metabolic measurements.

		First Trimester		Second Trimester		Third Trimester	
	*n*	r	r_adj_	r	r_adj_	r	r_adj_
Adiposity markers							
ppBMI	101	−0.40 ***	-	-	-	-	-
SAT thickness	84	−0.28 *	0.04	-	-	-	-
VAT thickness	84	−0.10	0.14	-	-	-	-
Leptin	74	−0.34 **	0.02	−0.34 **	−0.20 ^†^	−0.27 *	−0.10
Glucose homeostasis markers							
Fasting glucose	88	−0.09	0.20 ^†^	0.10	0.18 ^†^	−0.03	0.06
Fasting insulin	88	−0.21 *	0.14	−0.13	−0.02	−0.22 *	−0.09
HOMA-IR	88	−0.21 ^†^	0.17	−0.09	0.01	−0.18 ^†^	−0.05
Inflammatory markers							
Adiponectin	72	0.17	0.01	0.25 *	0.16	0.17	0.10
IL-6	70	−0.26 *	0.01	−0.28 *	−0.26 *	0.002	0.06
CRP	70	−0.29 *	−0.28 *	−0.07	0.08	0.03	0.15

Values are presented as Pearsonor Spearman correlation coefficients, raw (r) or adjusted for ppBMI (radj). *** *p* < 0.0001, ** *p* < 0.01, * *p* < 0.05, ^†^
*p* < 0.1. ppBMI, pre-pregnancy body mass index; SAT, subcutaneous adipose tissue; VAT, visceral adipose tissue; HOMA-IR, homeostasis model assessment of insulin resistance; IL-6, interleukin-6; CRP, C-reactive protein.

## Data Availability

The datasets generated and/or analyzed during the present study are available from the corresponding author upon reasonable request.

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
