# Peer review of "Trimester-Specific Serum Fructosamine in Association with Abdominal Adiposity, Insulin Resistance, and Inflammation in Healthy Pregnant Individuals"

_nutrients, 2022, doi:10.3390/nu14193999_

Round 1
Reviewer 1 Report
The manuscript ‘Trimester-Specific Serum Fructosamine in Association With Abdominal Adiposity, Insulin Resistance, and Inflammation in Healthy Pregnant Individuals’ is an exploratory analysis of two prospective studies, which aims to characterize the variations of fructosamine across trimesters and examine its associations with adiposity, glucose homeostasis and inflammation measurements. The findings demonstrated the trend of fructosamine concentration with the progress of pregnancy and the associations between fructosamine and adiposity, fasting insulin, HOMA-IR, as well as some inflammatory markers. The manuscript is generally well written but needs a major revision before publication.
The major concerns are:
(1) Introduction, Page 3-4, Line 91-102: a) the background of albumin-corrected fructosamine was not clear; b) the author did not show the importance of ppBMI as the main confounder; c) there was no enough evidence to support the hypothesis about the differential trends of uncorrected and corrected fructosamine levels.
(2) Materials and Methods, Page 5, Line 212: “Correlation coefficients are presented as both raw and adjusted for ppBMI.” Because the significance of most associations disappeared after the adjustment, stratified analysis according to ppBMI may be a useful supplement.
(3) The content in Discussion Section should be concentrated. Repeated statements were not necessary.
Page 10, Line 337-338: “To our knowledge, this is the first study to examine the associations between fructosamine concentrations and adiposity in the gestational context.”
Page 10, Line 355-357: To our knowledge, no other group has ever examined these associations in pregnancy and very few studies are available for comparison in the general population.”
Page 11, Line 394-396: “Because this is the first study to examine the associations between fructosamine and inflammation in pregnancy, no other report can be used as a comparator.”
Page 12, Line 444-446: “To the best of our knowledge, this is the first study to examine the associations between fructosamine concentrations and a wide diversity of metabolic measurements (adiposity, glucose homeostasis, and inflammation).”
Other minor comment:
(1) Table 1 and Figure 1: The unit of concentration of albumin-corrected fructosamine should be μmol/g, not μmol/L.
(2) The distributions of metabolic measurements are suggested to be presented in Table 1.
Reviewer 2 Report
This manuscript by Bernier et al. describes the longitudinal variations of serum fructosamine in a population of healthy pregnant women. The data point to significant associations between fructosamine levels and markers of adiposity at first trimester.
Significance
This is the first comprehensive study to analyze serum fructosamine levels in healthy pregnant women with a wide range of adiposity.
Specific comments
1-The introduction needs to be remodeled to better reflect the aims of the study as stated in the title:
- The rationale to examine the associations between circulating fructosamine levels and abdominal adiposity needs to be clearly supported. The point that measurements had never been done before in healthy pregnant women (line 90) does not make a strong enough case to implement a study.
- GDM was diagnosed at 24 weeks follow up, thus investigating the variations of serum fructosamine in pregnancy with GDM was not the primary aim of the study. I suggest to replace the first paragraph line 45 to 68 related to GDM with a background focusing on adiposity/obesity in pregnancy.
- Lines 73-79 belongs to discussion
2- The longitudinal raw data for VAT/SAT measures, leptin and BMI should be provided in the result section for each BMI group.
3- Compared with markers of insulin resistance, there is a much weaker association between fasting glucose and fructosamine levels. The results are in line with data reporting that second trimester fructosamine is a poor predictor of gestational glucose tolerance glycemic indices (22). These observations need to be better highlighted in the discussion section.
4-Because the small number of women with GDM lacks power in the association analysis, it may be wiser to delete the GDM group from the study in order to reach a straightforward conclusion relating to adiposity (see your statements lines 380, 455).
5. The sentence “Fructosamine refers to all glycated proteins in serum, with albumin being the main component [20,21]” lacks clarity. I guess the authors wanted to say that this is what the colorimetric test measures rather than what exactly is fructosamine.
6. Leptin is a marker of adiposity, not a marker of inflammation line 392. Please correct in table 3 and text.
7. What is the meaning of the sentence line 461 when considering a cohort of pregnant women ?
Author Response
RESPONSE TO REVIEWER 2
We would like to thank the Reviewer for their review of our manuscript. The comments and suggestions provided were taken into consideration and were incorporated into the revised manuscript, as detailed below. As a result, we believe that the merit of our manuscript has improved substantially.
1-The introduction needs to be remodeled to better reflect the aims of the study as stated in the title:
-The rationale to examine the associations between circulating fructosamine levels and abdominal adiposity needs to be clearly supported. The point that measurements had never been done before in healthy pregnant women (line 90) does not make a strong enough case to implement a study.
-GDM was diagnosed at 24 weeks follow up, thus investigating the variations of serum fructosamine in pregnancy with GDM was not the primary aim of the study. I suggest to replace the first paragraph line 45 to 68 related to GDM with a background focusing on adiposity/obesity in pregnancy.
Response: We do agree that the introduction do not accurately reflets the main objectives and title of the study. As you correctly mentioned, GDM was diagnosed after several weeks of follow-ups (on average between 24 and 28 weeks) and the investigation of serum fructosamine variations during pregnancy with GDM was not the main purpose of the study. Although, we still find it necessary to support our introduction with existing literature on the reasons supporting the potential use of fructosamine as a short-term glycemic marker, which was mostly being carried out in studies with populations with GDM. Still, significant changes were made accordingly, as the first paragraph (lines 47 to 66) was remodeled to include more background information regarding adiposity in the gestational context (see lines below).
Lines 47 to 66: Gestational diabetes mellitus (GDM) is one of the most frequent health condition affecting pregnant individuals [1]. This could partially be brought on by the fact that an increasing number of individuals enter pregnancy being overweight or obese, with increased adipose fat storage. Indeed, independently of other known risk factors for GDM, ultrasound-estimated maternal visceral adipose tissue (VAT) thickness in pregnancy is strongly associated with insulin resistance and higher risks of developing GDM [2-5]. Along with increased adiposity, GDM can have detrimental health consequences for both the mothers and the fetus [6-9]. Hence, glycemic control is particularly fundamental in all pregnant individuals (with or without GDM), as continuous associations between maternal glucose levels (below those of GDM diagnosis) and adverse pregnancy outcomes, such as primary cesarean delivery and neonatal hypoglycemia, have been observed [10]. Several risk factors have been implicated in the development of glucose intolerance and GDM, with maternal weight status being the most commonly evaluated reversible risk factor [9,11].
- Lines 73-79 belongs to discussion.
Response: As requested by Reviewer 1, this section of the introduction was improved (see lines below). We kept this section in the introduction, as it provides background information on the use of albumin-corrected fructosamine.
Lines 83-96: Nevertheless, conditions that affect serum protein and albumin metabolism can also influence fructosamine levels, as fructosamine levels correlate significantly with albumin and total protein concentrations [15,24]. Some methods are recommended in the literature to correct fructosamine levels based on total circulating protein or albumin, as corrected fructosamine is shown to better correlate with glucose homeostasis markers and GDM diagnosis [24]. Since albumin concentrations are known to decrease during pregnancy due to dilution [25,26] and to the natural low-grade inflammation developing over trimesters, correction for albumin concentrations could reverse the results observed with uncorrected fructosamine levels, as seen by others [27]. There is a debate about whether fructosamine concentration should be corrected for total albumin, total protein, or neither, and best practices have not been established [28]. Therefore, in this study, we use both uncorrected and albumin-corrected fructosamine levels, thus enabling comparison with other studies.
- The longitudinal raw data for VAT/SAT measures, leptin and BMI should be provided in the result section for each BMI group.
Response: In response to this comment, longitudinal raw data for VAT/SAT measures, leptin and ppBMI, for each ppBMI categories were added as supplementary files (Table S1). We also included the longitudinal raw data for all other metabolic markers. A statement was also added under Table 1 (see lines below). Since it did not directly respond to the study’s main objectives, we preferred to only present those results in supplementary data. Moreover, our team recently publish an article aiming to quantify changes in circulating concentrations of leptin, adiponectin, IL-6 and CRP across trimesters of pregnancy, according to ppBMI (Savard et al., Appl Physiol Nutr Metab. 2022).
Line 282: Longitudinal raw data for every metabolic marker according to ppBMI categories are available in Supplementary materials (Table S1).
- Compared with markers of insulin resistance, there is a much weaker association between fasting glucose and fructosamine levels. The results are in line with data reporting that second trimester fructosamine is a poor predictor of gestational glucose tolerance glycemic indices (22). These observations need to be better highlighted in the discussion section.
Response: We agree and added a statement in the discussion.
Lines 452-429: Finally, it should be noted that much weaker correlations were observed between fasting glucose and fructosamine levels than between markers of insulin resistance and fructosamine levels. Those findings are consistent with evidence suggesting that fructosamine concentrations are a poor predictor of gestational glucose tolerance and risk of developing GDM [31,34,35].
- Because the small number of women with GDM lacks power in the association analysis, it may be wiser to delete the GDM group from the study in order to reach a straightforward conclusion relating to adiposity (see your statements lines 380, 455)
Response: In response to this comment, we conducted all analyses without participants with GDM and found similar results. Also, since GDM is usually diagnosed between the 24th and 28th weeks of gestation and may not have any influence on metabolic markers in the first trimester, we consider that data of those individuals before and even after GDM diagnosis (as they can be normalized with therapy), should be kept in our study.
- The sentence “Fructosamine refers to all glycated proteins in serum, with albumin being the main component [20,21]” lacks clarity. I guess the authors wanted to say that this is what the colorimetric test measures rather than what exactly is fructosamine.
Response: We agree and changes were made accordingly (see lines below).
Lines 76-79: Fructosamine is a sugar-protein complex that forms in chronic hyperglycemic conditions. Since albumin is the most abundant protein in the blood, fructosamine levels typically reflect albumin glycation [20,21].
- Leptin is a marker of adiposity, not a marker of inflammation line 392. Please correct in table 3 and text.
Response: The requested corrections were made as leptin is now presented as an adiposity marker, along with ppBMI, subcutaneous, and visceral adipose tissue thicknesses, both in the text and tables.
- What is the meaning of the sentence line 461 when considering a cohort of pregnant women ?
Response: In the spirit of equity, diversity and inclusion, this sentence was added to reflect the missing information about gender in this cohort. Recognizing that diverse groups of people who do not identify as women have perinatal needs and experiences that may be similar to but also unique from those of cisgender women, we did not want to assume that all recruited individuals identify as women, as it could include pregnant individuals of all genders.
Note to the Editors : Please note that I add problems with the system while responding to Reviewer 1 , as I was not able to attached any document. Therefore, I attached in this form the response intended for Reviewer 1 (please see the attachement). I tried to reach to your Technical Assistance Service, but no one seemed available. Thank you for your understanding.
Round 2
Reviewer 1 Report
Dear editor,
I have no further comments to the authors.